# SGD AND WEIGHT DECAY PROVABLY INDUCE A LOW-RANK BIAS IN NEURAL NETWORKS

## ABSTRACT

We analyze deep ReLU neural networks trained with mini-batch Stochastic Gradient Descent (SGD) and weight decay. We show, both theoretically and empirically, that when training a neural network using SGD with weight decay and small batch size, the resulting weight matrices tend to be of small rank. Our analysis relies on a minimal set of assumptions; the neural networks may be arbitrarily wide or deep, and may include residual connections, as well as convolutional layers. The same analysis implies the inherent presence of SGD "noise", defined as the inability of SGD to converge to a stationary point. In particular, we prove that SGD noise must always be present, even asymptotically, as long as we incorporate weight decay and the batch size is smaller than the total number of training samples.

## 1   INTRODUCTION

Stochastic gradient descent (SGD) is one of the standard workhorses for optimizing deep models (Bottou, 1991). Though initially proposed to remedy the computational bottleneck of gradient descent (GD), recent studies suggest that SGD also induces crucial regularization, which prevents overparameterized models from converging to minima that cannot generalize well (Zhang et al., 2016; Jastrzebski et al., 2017; Keskar et al., 2017; Zhu et al., 2019). Empirical studies suggest that SGD outperforms GD Zhu et al. (2019) and SGD generalizes better when used with smaller batch sizes (Hoffer et al., 2017; Keskar et al., 2017), and (iii) gradient descent with additional noise cannot compete with SGD Zhu et al. (2019). The full range of regularization effects induced by SGD, however, is not yet fully understood.

In this paper we present a mathematical analysis of the bias of SGD towards rank-minimization. To investigate this bias, we propose the *SGD Near-Convergence Regime* as a novel approach for investigating inductive biases of SGD-trained neural networks. This setting considers the case where SGD reaches a point in training where the expected update is small in comparison to the actual weights' norm. Our analysis is fairly generic: we consider deep ReLU networks trained with mini-batch SGD for minimizing a differentiable loss function with $L_2$ regularization (i.e., weight decay). The neural networks may include fully-connected layers, residual connections and convolutions.

Our main contributions are:

- In Thm. 1, we demonstrate that training neural networks with mini-batch SGD and weight decay results in a low-rank bias in their weight matrices. We theoretically demonstrate that when training with smaller batch sizes, the rank of the learned matrices tends to decrease. This observation is validated as part of an extensive empirical study of the effect of certain hyperparameters on the rank of learned matrices with various architectures.

- In Sec. 3.2, we study the inherent inability of SGD to converge to a stationary point, that we call 'SGD noise'. In Props. 1-2 we describe conditions in which 'SGD noise' is inevitable when training convolutional neural networks. In particular, we demonstrate that when training a fully-connected neural network, SGD noise must always be present, even asymptotically, as long as we incorporate weight decay and the batch size is smaller than the total number of samples. These predictions are empirically validated in Sec. 4.3.

## 1.1 RELATED WORK

A prominent thread in the recent literature revolves around characterizing the implicit regularization of gradient-based optimization in the belief that this is key to generalization in deep learning. Several papers have focused on a potential bias of gradient descent or stochastic gradient descent towards rank minimization. The initial interest was motivated by the matrix factorization problem, which corresponds to training a depth-2 linear neural network with multiple outputs w.r.t. the square loss. Gunasekar et al. (2017) initially conjectured that the implicit regularization in matrix factorization can be characterized in terms of the nuclear norm of the corresponding linear predictor. This conjecture, however, was formally refuted by Li et al. (2020). Later, Razin & Cohen (2020) conjectured that the implicit regularization in matrix factorization can be explained by rank minimization, and also hypothesized that some notion of rank minimization may be key to explaining generalization in deep learning. Li et al. (2020) established evidence that the implicit regularization in matrix factorization is a heuristic for rank minimization. Beyond factorization problems, Ji & Telgarsky (2020) showed that gradient flow (GF) training of univariate linear networks w.r.t. exponentially-tailed classification losses learns weight matrices of rank 1.

With nonlinear neural networks, however, things are less clear. Empirically, a series of papers (Denton et al., 2014; Alvarez & Salzmann, 2017; Tukan et al., 2021; Yu et al., 2017; Arora et al., 2018) showed that replacing the weight matrices by low-rank approximations results in only a small drop in accuracy. This suggests that the weight matrices at convergence may be close to low-rank matrices. However, whether they provably behave this way remains unclear. Timor et al. (2022) showed that for ReLU networks, GF generally does not minimize rank. They also argued that sufficiently deep ReLU networks can have low-rank solutions under $L_2$ norm minimization. This interesting result, however, applies to layers added to a network that already solves the problem and may not have any low-rank bias. It is not directly related to the mechanism described in this paper, which applies to all layers in the network but only in the presence of regularization and SGD unlike (Timor et al., 2022). A recent paper (Le & Jegelka, 2022) analyzes low-rank bias in neural networks trained with GF (without regularization). While this paper makes significant strides in extending the analysis in (Ji & Telgarsky, 2020), it makes several limiting assumptions. As a result, their analysis is only applicable under very specific conditions, such as when the data is linearly separable, and their low-rank analysis is limited to a set of linear layers aggregated at the top of the trained network.

## 2 PROBLEM SETUP

In this work we consider a standard supervised learning setting (classification or regression), and study the inductive biases induced by training a neural network with mini-batch SGD along with weight decay. Formally, the task is defined by a distribution $P$ over samples $(x, y) \in \mathcal{X} \times \mathcal{Y}$, where $\mathcal{X} \subset \mathbb{R}^{c_1 \times h_1 \times w_1}$ is the instance space (e.g., images), and $\mathcal{Y} \subset \mathbb{R}^k$ is a label space.

We consider a parametric model $\mathcal{F} \subset \{f' : \mathcal{X} \to \mathbb{R}^k\}$, where each function $f_W \in \mathcal{F}$ is specified by a vector of parameters $W \in \mathbb{R}^N$. A function $f_W \in \mathcal{F}$ assigns a prediction to any input point $x \in \mathcal{X}$, and its performance is measured by the *Expected Risk*, $L_P(f_W) := \mathbb{E}_{(x,y) \sim P}[\ell(f_W(x), y)]$, where $\ell : \mathbb{R}^k \times \mathcal{Y} \to [0, \infty)$ is a non-negative, differentiable, loss function (e.g., MSE or cross-entropy losses). For simplicity, in the analysis we focus on the case where $k = 1$.

Since we do not have direct access to the full population distribution $P$, the goal is to learn a predictor, $f_W$, from some training dataset $S = \{(x_i, y_i)\}_{i=1}^m$ of independent and identically distributed (i.i.d.) samples drawn from $P$. Traditionally, in order to avoid overfitting the training data, we typically employ weight decay in order to control the complexity of the learned model. Namely, we intend to minimize the *Regularized Empirical Risk*, $L_S^\lambda(f_W) := \frac{1}{m} \sum_{i=1}^m \ell(f_W(x_i), y_i) + \lambda \|W\|_2^2$, where $\lambda > 0$ is predefined hyperparameter. In order to minimize this objective, we typically use mini-batch SGD, as detailed below.

**Optimization.** In this work, we minimize the regularized empirical risk $L_S^\lambda(f_W)$ by applying stochastic gradient descent (SGD) for a certain number of iterations $T$. Formally, we initialize $W_0$ using a standard initialization procedure, iteratively update $W_t$ for $T$ iterations and return $W_T$. At each iteration, we sample a subset $\tilde{S} = \{(x_{ij}, y_{ij})\}_{j=1}^B \subset S$ uniformly at random and update $W_{t+1} \leftarrow W_t - \mu \nabla_W L_{\tilde{S}}^\lambda(f_{W_t})$, where $\mu > 0$ is a predefined learning rate.

**Notation.** Throughout the paper, we use the following notations. For an integer $k \geq 1$, $[k] = \{1, \ldots, k\}$. $\|z\|$ denotes the Euclidean norm. For two given vectors $x \in \mathbb{R}^n, y \in \mathbb{R}^m$ we define their concatenation as follows $(x\|y) := (x_1, \ldots, x_n, y_1, \ldots, y_m) \in \mathbb{R}^{n+m}$. For a given matrix $x \in \mathbb{R}^{n \times m}$, we denote $x_{i,:}$ its $i$'th row and by $\mathrm{vec}(x) := (x_1\|\ldots\|x_n)$ its vectorization. For a given tensor $x \in \mathbb{R}^{c \times h \times w}$, we denote by $\mathrm{vec}(x) := (\mathrm{vec}(x_1)\|\ldots\|\mathrm{vec}(x_c))$ the vectorized form of $x$. We define tensor slicing as follows, $x_{a:b} := (x_a, \ldots, x_b)$.

## 2.1 ARCHITECTURES

In this work, the function $f_W$ represents a neural network, consisting of a set of layers of weights interlaced with ReLU activation units. We employ a fairly generic definition of a neural network, that includes convolutional layers, pooling layers, residual connections and fully-connected layers.

**Network architecture.** Formally, $f_W$ is as a directed acyclic graph (DAG) $G = (V, E)$, where $V = \{v_1, \ldots, v_L\}$ consists of the various layers of the network and each edge $e_{ij} = (v_i, v_j) \in E$ specifies a connection between two layers. Each layer is a function $v_i : \mathbb{R}^{c_1 \times h_1 \times w_1} \to \mathbb{R}^{c_i \times h_i \times w_i}$ and each connection $(v_i, v_j)$ holds a transformation $C^{ij} : \mathbb{R}^{c_j \times h_j \times w_j} \to \mathbb{R}^{c_i \times h_i \times w_i}$. The layers are divided into three categories: **(i)** the input layer $v_1$, **(ii)** the output layer $v_L$ and **(iii)** intermediate layers. In this setting, we do not have connections directed towards the input layer nor connections coming out of the output layer (i.e., $\forall\, i \in [L] : (v_L, v_i), (v_0, v_i) \notin E$). Given an input $x \in \mathbb{R}^{c_i \times h_i \times w_i}$, the output of a given layer $v_i$ is evaluated as follows $v_i(x) := \sigma(\sum_{j \in \mathrm{pred}(i)} C^{ij}(v_j(x)))$, except for the output layer $v_L$ that computes $f_W(x) := v_L(x) := \sum_{j \in \mathrm{pred}(L)} C^{Lj}(v_j(x))$. Here, $\mathrm{pred}(i) := \{j \in [L] \mid (v_i, v_j) \in E\}$, $\mathrm{succ}(i) := \{j \in [L] \mid (v_j, v_i) \in E\}$ and $\sigma$ is the ReLU activation function. Each transformation $C^{ij}$ is either trainable (e.g., a convolutional layer) or a constant affine transformation (e.g., a residual connection). We denote by $E_T$ the set of trainable connections. In this paper, we consider the following transformations.

**Convolutional layers.** A convolutional layer (Lecun et al., 1998) (see also Goodfellow et al. (2016)) $C^{ij} : \mathbb{R}^{c_j \times h_j \times w_j} \to \mathbb{R}^{c_i \times h_i \times w_i}$ with kernel sizes $(k_1, k_2)$, padding $p$ and stride $s$ is parameterized by a tensor $Z^{ij} \in \mathbb{R}^{c_i \times c_j \times k_1 \times k_2}$ and computes the following tensor as output

$$\forall\,(c,t,l) \in [c_i] \times [h_i] \times [w_i] : \; y_{c,t,l} = \sum_{c'=1}^{c_j} \mathrm{vec}(Z_{c,c',:,:})^\top \cdot \mathrm{vec}(\mathrm{Pad}_p(x)_{c',ts\,:\,(t+1)s+k_1, ls\,:\,(l+1)s+k_2}). \quad (1)$$

Here, $\mathrm{Pad}_p$ takes a tensor $x \in \mathbb{R}^{c \times h \times w}$ and returns a new tensor $x' \in \mathbb{R}^{c \times (h+2p) \times (w+2p)}$, where the first and last $p$ rows and columns of each channel $x'_{c,\,:\,,\,:}$ are zeros and the middle $1 \times h \times w$ tensor is equal to $x_{c,\,:\,,\,:}$. The formulas for the output dimensions are $h_2 = (\lfloor (h_1 - k_1 + 2p) \rfloor / s + 1)$ and $w_2 = (\lfloor (w_1 - k_2 + 2p) \rfloor / s + 1)$. For a given convolutional layer $C^{ij}$ with weights $Z^{ij}$, we define a matrix $V^{ij} \in \mathbb{R}^{c_i h_i w_i \times c_j h_j w_j}$ that computes $\forall\, x \in \mathbb{R}^{c_j \times h_j \times w_j} : V^{ij} \mathrm{vec}(x) = \mathrm{vec}(C^{ij}(x))$. This matrix exists since both the padding and convolution operations can be represented as linear operations of the input. We also consider $W^{ij} \in \mathbb{R}^{c_i \times c_j k_1 k_2}$ which is a matrix whose $c$'th row is the vectorized filter $W_c^{ij} := \mathrm{vec}(Z_{c,\,:\,,\,:}^{ij})$.

**Fully-connected layers.** As a special case of convolutional layers, the network may also include fully-connected layers. Namely, a fully-connected layer $F : \mathbb{R}^{c_1} \to \mathbb{R}^{c_2}$ associated with a matrix $W \in \mathbb{R}^{c_2 \times c_1}$ can be represented as a $1 \times 1$ convolutional layer $C : \mathbb{R}^{c_1 \times 1 \times 1} \to \mathbb{R}^{c_2 \times 1 \times 1}$ with $k_1 = k_2 = 1$, $p = 0$ and $s = 1$. Namely, the parameters tensor $Z \in \mathbb{R}^{c_2 \times c_1 \times 1 \times 1}$ satisfies $Z_{a,b,1,1} = W_{a,b}$ for all $(a, b) \in [c_2] \times [c_1]$ and the layer satisfies $\mathrm{vec}(C(x)) = W\mathrm{vec}(x)$.

**Pooling layers.** A pooling layer (Zhou & Chellappa, 1988) (see also (Goodfellow et al., 2016)) $C$ with kernel dimensions $(k_1, k_2)$ stride $s$ and padding $p$ takes an input $x \in \mathbb{R}^{c_1 \times h_1 \times w_1}$ and computes an output $y \in \mathbb{R}^{c_2 \times h_2 \times w_2}$ with $c_2 = c_1$ channels, and dimensions $h_2 = (\lfloor (h_1 - k_1 + 2p) \rfloor / s + 1)$ and $w_2 = (\lfloor (w_1 - k_2 + 2p) \rfloor / s + 1)$. Each pooling layer computes the following tensor as output

$$\forall\,(c,t,l) \in [c_i] \times [h_i] \times [w_i] : \; y_{c,t,l} = \mathrm{op}(\mathrm{Pad}_p(x)_{c,ts\,:\,(t+1)s+k_1, ls\,:\,(l+1)s+k_2}), \quad (2)$$

where op is either the maximum or average operator.

**Rearrangement layers.** To conveniently switch between convolutional and fully-connected layers, we should be able to represent tensor layers as vectors and vice versa. To reshape the representation of a certain layer, we allow the networks to include rearrangement layers. A rearrangement layer

$C^{ij} : \mathbb{R}^{c_j \times h_j \times w_j} \to \mathbb{R}^{c_i \times h_i \times w_i}$ takes an input vector $x \in \mathbb{R}^{c_j \times h_j \times w_j}$ and 'rearranges' its coordinates in a different shape and permutation. Formally, it returns a vector $(x_{\pi(k)})_{k \in [c_j] \times [h_j] \times [w_j]}$, where $\pi : [c_j] \times [h_j] \times [w_j] \to [c_i] \times [h_i] \times [w_i]$ is invertible (in particular, $c_i h_i w_i = c_j h_j w_j$).

## 3 THEORETICAL RESULTS

In this section we describe our main theoretical results. We investigate the inductive biases that emerge near convergence when training with SGD. For this purpose, we begin by introducing our definitions of SGD convergence (and near convergence) points.

To formally study convergence, we employ the notion of convergence in mean of random variables. Namely, we say that a sequence of random variables $\{W_t\}_{t=1}^{\infty} \subset \mathbb{R}^N$ starting from a certain vector $W_0 \in \mathbb{R}^N$ (constant) converges in mean if the following holds

$$\exists\, W^* \in \mathbb{R}^N : \lim_{t \to \infty} \mathbb{E}[\|W_t - W^*\|] = 0.$$

As a consequence of this definition, we have

$$\lim_{t \to \infty} \mathbb{E}[\|W_{t+1} - W_t\|] \leq \lim_{t \to \infty} \mathbb{E}[\|W_{t+1} - W^*\|] + \lim_{t \to \infty} \mathbb{E}[\|W_t - W^*\|] = 0,$$

where the expectations are taken over the selections of the mini-batches. In particular, convergence is possible only when the expected size of each step tends to zero. In particular, when training the network using mini-batch SGD along with weight decay, we have

$$\lim_{t \to \infty} \mathbb{E}_{\tilde{S}}[\|\nabla L_{\tilde{S}}^{\lambda}(f_{W_t})\|] = \lim_{t \to \infty} \tfrac{1}{\mu} \mathbb{E}[\|W_{t+1} - W_t\|] = 0. \tag{3}$$

In this work we study the implicit biases of SGD by investigating the *"SGD Near Convergence Regime"*, where SGD arrives at a point in training where each subsequent step is small compared to the actual weights, i.e., $\mathbb{E}_{\tilde{S}}[\|\nabla_{W^{ij}} L_{\tilde{S}}^{\lambda}(f_{W_T})\| / \|W_T^{ij}\|]$ is small. In Sec. 3.1 we show that near-convergence, mini-batch SGD learns neural networks with low-rank matrices and in Sec. 3.2 we prove that perfect SGD convergence is impossible in the presence of weight decay.

We acknowledge that our definition differs from the traditional notions of convergence. Several papers study the convergence of SGD to a point where the performance is near-optimal when the learning rate $\mu_t$ decays with the number of iterations. In these cases, convergence is guaranteed since $\mu_t$ tends to zero. These papers, however, do not analyze whether the expected gradients $\mathbb{E}_{\tilde{S}}[\|\nabla_{W_t^{ij}} L_{\tilde{S}}^{\lambda}(f_{W_t})\|]$ also decays at each step. Other papers (e.g., (Soudry & Carmon, 2016; Cooper, 2021)) study the critical points of the objective function to better understand the solutions of gradient-based optimization methods. While understanding the critical points of the objective function is necessary for characterizing the convergence points of GD and GF, SGD does not necessarily converge at these points. For instance, suppose $W_t$ is a stationary point of $L_S^{\lambda}(f_W)$ and there exists a batch $\tilde{S}$ for which $\nabla_W L_{\tilde{S}}^{\lambda}(f_{W_t}) \neq 0$. With probability $\binom{m}{B}^{-1} > 0$ SGD selects the batch $\tilde{S}$ and updates $W_{t+1} = W_t - \mu \nabla L_{\tilde{S}}^{\lambda}(f_{W_t}) \neq W_t$. Therefore, $p = \mathbb{P}[W_{t+1} \neq W_t \mid W_t] > 0$ and $\mathbb{P}[\exists\, l \in [T'] : W_{t+l} \neq W_t \mid W_t] \geq 1 - (1-p)^{T'} \xrightarrow[T' \to \infty]{} 1$. As a result, the probability of the optimization becoming stuck at $W_t$ indefinitely is zero.

### 3.1 LOW-RANK BIAS IN NEURAL NETWORKS

We begin our theoretical analysis with the simple observation (proved in Appendix A) that the number of input patches $N^{ij}$ of a certain convolutional layer $C^{ij}$ upper bounds the rank of the gradient of the network w.r.t. $W^{ij}$.

**Lemma 1.** *Let $f_W$ be a neural network and let $C^{ij}$ be a convolutional layer within $f_W$ with parameters matrix $W^{ij}$. Then,* $\text{rank}(\nabla_{W^{ij}} f_W(x)) \leq N^{ij}$.

Interestingly, we obtain particularly degenerate gradients for fully-connected layers. As discussed in Sec. 2.1 for a fully-connected layer $C^{ij} : \mathbb{R}^{c_j \times 1 \times 1} \to \mathbb{R}^{c_i \times 1 \times 1}$ we have $N^{ij} = 1$, and therefore, $\text{rank}(\nabla_{W^{ij}} f_W(x)) \leq 1$.

The following theorem provides an upper bound on the minimal distance between the network's weight matrices and low-rank matrices.

**Theorem 1.** *Let $\| \cdot \|$ be any matrix norm and $\ell$ any differentiable loss function. Let $f_W(x)$ be a ReLU neural network and $C^{ij}$ be a convolutional layer within $f_W$ and let $B \in [m]$. Then,*

$$\min_{\substack{W \in R^{d_i \times d_j}: \\ \mathrm{rank}(W) \leq N^{ij}B}} \|\tfrac{W^{ij}}{\|W^{ij}\|} - W\| \ \leq \ \tfrac{1}{2\lambda} \min_{\substack{\tilde{S} \subset S: \\ |\tilde{S}|=B}} \|\nabla_{W^{ij}} L_{\tilde{S}}^{\lambda}(f_W)\| / \|W^{ij}\|$$

*Proof.* Let $\tilde{S} \subset S$ be a batch of size $B$. By the chain rule, we can write the gradient of the loss function as follows

$$\nabla_{W^{ij}} L_{\tilde{S}}^{\lambda}(f_W) \ = \ \tfrac{1}{B} \sum_{(x,y) \in \tilde{S}} \tfrac{\partial \ell(f_W(x),y)}{\partial f_W(x)} \cdot \nabla_{W^{ij}} f_W(x) + 2\lambda W^{ij} =: -E_{\tilde{S}} + 2\lambda W^{ij}.$$

According to Lem. 1, we have $\mathrm{rank}(\tfrac{1}{2\lambda} E_{\tilde{S}}) \leq BN^{ij}$. Therefore, we obtain that

$$\min_{W: \, \mathrm{rank}(W) \leq BN^{ij}} \|W^{ij} - W\| \ \leq \ \min_{\tilde{S} \subset S: \, |\tilde{S}|=B} \|W^{ij} - \tfrac{1}{2\lambda} E_{\tilde{S}}\| \ = \ \tfrac{1}{2\lambda} \min_{\tilde{S} \subset S: \, |\tilde{S}|=B} \left\|\nabla_{W^{ij}} L_{\tilde{S}}^{\lambda}(f_W)\right\|.$$

Finally, by dividing both sides by $\|W^{ij}\|$ we obtain the desired inequality. $\qquad\square$

The theorem above provides an upper bound on the minimal distance between the parameters matrix $W^{ij}$ and a matrix of rank $\leq \ BN^{ij}$. The upper bound is proportional to, $\min_{\tilde{S}} \|\nabla_{W^{ij}} L_{\tilde{S}}^{\lambda}(f_W)\| / \|W^{ij}\|$, which is the minimal norm of the gradient of the regularized empirical risk evaluated on batches of size $B$, normalized by the norm of the weight matrix. As shown in equation 3, near convergence we expect $\mathbb{E}_{\tilde{S}}[\|\nabla_{W^{ij}} L_{\tilde{S}}^{\lambda}(f_W)\| / \|W^{ij}\|]$ to be small, and therefore, $\min_{\tilde{S}} \|\nabla_{W^{ij}} L_{\tilde{S}}^{\lambda}(f_W)\| / \|W^{ij}\|$ should also be small ($\tilde{S}$ is distributed uniformly as a batch of samples size $B$). In particular, by the theorem above, we expect $\min_{W: \, \mathrm{rank}(W) \leq BN^{ij}} \|\tfrac{W^{ij}}{\|W^{ij}\|} - W\|$ to also be small. As a result, we predict that the rank of the learned parameter matrices $W^{ij}$ decreases with the batch size. In Sec. 4 we validate this idea with an extensive set of experiments – and also study the relationship between the rank and other hyperparameters.

## 3.2 Degeneracy and the Origin of "SGD Noise"

As we mentioned, it is impossible for SGD to converge to a stationary point of the gradient dynamical system, resulting in inherent "SGD noise". In this section, we study the (non-)convergence of mini-batch SGD. Our results are essentially impossibility results: the assumption of SGD convergence to a critical point of the gradient implies that the network represents the zero function. As a result, asymptotic noise is inherently unavoidable (when training properly). For simplicity, we assume that $\forall \, i \in [m]: \ x_i \neq 0$.

As shown in equation 3, any convergence point $W$ of SGD satisfies $\mathbb{E}_{\tilde{S}}[\|\nabla L_{\tilde{S}}^{\lambda}(f_W)\|] = 0$. Since the distribution over mini-batches $\tilde{S}$ of size $B$ is discrete and $\|\nabla L_{\tilde{S}}^{\lambda}(f_W)\| \geq 0$, we obtain that at convergence we have $\|\nabla_{W^{ij}} L_{\tilde{S}}^{\lambda}(f_W)\| = 0$, for all mini-batches $\tilde{S}$ of size $B$. In particular,

$$\forall \tilde{S}: \ 0 \ = \ \nabla_{W^{ij}} L_{\tilde{S}}^{\lambda}(f_W) \ = \ \frac{1}{B} \sum_{(x,y) \in \tilde{S}} \frac{\partial \ell(f_W(x),y)}{\partial f_W(x)} \cdot \nabla_{W^{ij}} f_W(x) + 2\lambda W^{ij}. \tag{4}$$

Suppose we have two batches $\tilde{S}_1, \tilde{S}_2 \subset S$ of size $B$ that differ by only one sample. We denote the unique sample of each batch by $(x_{j_1}, y_{j_1})$ and $(x_{j_2}, y_{j_2})$ respectively. We notice that,

$$0 \ = \ \nabla_{W^{ij}} L_{\tilde{S}_1}^{\lambda}(f_W) - \nabla_{W^{ij}} L_{\tilde{S}_2}^{\lambda}(f_W)$$

$$= \ \frac{\partial \ell(f_W(x_{j_1}),y_{j_1})}{\partial f_W(x_{j_1})} \cdot \nabla_{W^{ij}} f_W(x_{j_1}) - \frac{\partial \ell(f_W(x_{j_2}),y_{j_2})}{\partial f_W(x_{j_2})} \cdot \nabla_{W^{ij}} f_W(x_{j_2}).$$

Therefore, we conclude that for all $j_1, j_2 \in [m]$,

$$M^{ij} \ = \ \frac{\partial \ell(f_W(x_{j_1}),y_{j_1})}{\partial f_W(x_{j_1})} \cdot \nabla_{W^{ij}} f_W(x_{j_1}) \ = \ \frac{\partial \ell(f_W(x_{j_2}),y_{j_2})}{\partial f_W(x_{j_2})} \cdot \nabla_{W^{ij}} f_W(x_{j_2}). \tag{5}$$

Hence, for all $(v_i, v_j) \in E_T$ and $k \in [m]$,

$$\frac{\partial \ell(f_W(x_k),y_k)}{\partial f_W(x_k)} \cdot \nabla_{W^{ij}} f(x_k) + 2\lambda W^{ij} \ = \ M^{ij} + 2\lambda W^{ij} \ = \ 0. \tag{6}$$

Therefore, unless $\lambda = 0$ or $\forall\, (v_i, v_j) \in E_T : W^{ij} = 0$, we conclude that $\frac{\partial \ell(f_W(x_k), y_k)}{\partial f_W(x_k)} \neq 0$ for all $k \in [m]$. In this case, we also obtain that $\{\frac{\partial f_W(x_k)}{\partial \mathrm{vec}(W^{ij})}\}_{k=1}^m$ are collinear vectors by equation 5.

Therefore, any convergence point of training a neural network using mini-batch SGD along with weight decay is highly degenerate and does not fit any one of the training labels. To better understand the essence of this degeneracy, we provide the following proposition (proved in Appendix A), which is specialized for ReLU networks.

**Proposition 1** ($\lambda > 0$). *Let $\ell(a, b)$ be a differentiable loss function, $\lambda > 0$, and let $f_W(x)$ be a ReLU neural network, where $\mathrm{succ}(1) = \{p\}$ and $(v_p, v_1) \in E_T$. Let $\{x_i^k\}_{k=1}^{N^{p1}}$ be the $N^{p1}$ patches of $x_i$ used by the layer $C^{p1}$. Let $W$ be a convergence point of mini-batch SGD for minimizing $L_S^\lambda(f_W)$ (see equation 4). Then, either $f_W \equiv 0$ or $\forall\, i, j \in [m] : \{x_i^k, x_j^k\}_{k=1}^{N^{p1}}$ are linearly dependent tensors.*

The preceding proposition shows that unless the patches of any two training samples are linearly dependent, any convergence point of SGD corresponds to the zero function. When $N^{p1}$ is small, the linear dependence criterion is unrealistic. For example, if $C^{p1}$ is a fully-connected layer, $N^{p1} = 1$, and the condition asserts that any two training samples $x_i, x_j$ are collinear. Since this is unrealistic, we conclude that convergence is impossible unless $f_W \equiv 0$.

As a next step, we consider convergence of SGD when training without weight decay.

**Proposition 2** ($\lambda = 0$). *Let $\lambda = 0$ and let $\ell$ be a differentiable loss function. Let $f_W(x)$ be a ReLU neural network, where $\mathrm{succ}(1) = \{p\}$ and $(v_p, v_1) \in E_T$ is fully-connected. Let $\{x_i^k\}_{k=1}^{N^{p1}}$ be the $N^{p1}$ patches of $x_i$ used by the layer $C^{p1}$. Let $W$ be a convergence point of mini-batch SGD for minimizing $L_S^\lambda(f_W)$. Then, $\forall\, i \in [m] : \frac{\partial \ell(f_W(x_i), y_i)}{\partial f_W(x_i)} = 0$ or $f_W(x_j) = 0$ or $\forall\, j \in [m] : \{x_i^k, x_j^k\}_{k=1}^{N^{p1}}$ are linearly dependent tensors.*

The preceding proposition (proved in Appendix A) provides conditions for SGD convergence of training a convolutional network without weight decay. It shows that at convergence, for every sample the network either perfectly fits the label (i.e., $\frac{\partial \ell(f_W(x_i), y_i)}{\partial f_W(x_i)} = 0$) or outputs zero unless the patches of that sample are linearly dependent with the patches of any other sample. As mentioned, the linear dependence criteria is generally unrealistic when $(v_p, v_1)$ is fully-connected. Therefore, convergence with fully-connected networks is possible only when $f_W$ perfectly fits the training set labels.

We note that if $\ell(a, b)$ is convex and has no minima $a$ for any $b \in \mathbb{R}$ (e.g., binary cross-entropy, logistic loss or exponential loss), then $\forall\, i \in [m] : \frac{\partial \ell(f_W(x_i), y_i)}{\partial f_W(x_i)} \neq 0$. Therefore, the only possible convergence points of fully-connected networks are ones for which $\forall\, i \in [m] : f_W(x_i) = 0$. Since this is in general absurd, we argue that perfect convergence of training a network with exponential-type loss functions is generally impossible. While convergence to a non-zero function is not guaranteed, in practice training without weight decay may still fall into the regime of 'almost convergence', in which $\max_{i \in [m]} \left| \frac{\partial \ell(f_W(x_i), y_i)}{\partial f_W(x_i)} \right|$ is tiny and as a result the training steps $-\mu \cdot \frac{\partial \ell(f_W(x_i), y_i)}{\partial f_W(x_i)} \cdot \frac{\partial f_W(x_i)}{\partial W^{ij}}$ are very small. This is usually the case when an overparameterized network has been properly trained. For the squared loss, convergence may occur when the network perfectly fits the training labels, i.e., $\forall\, i \in [m] : f_W(x_i) = y_i$.

It is worth noting that training with mini-batches during optimization and weight decay are critical to our analysis. While many papers examine the training dynamics and critical points of GD, as we show, SGD convergence points are highly degenerate and, in general, behave differently than GD solutions. Surprisingly, this analysis is unaffected by batch size, and therefore, the presence of SGD noise occurs regardless of batch size, as long as it is strictly smaller than the full dataset's size.

## 4 EXPERIMENTS

In this section we empirically study the implicit bias towards rank minimization in deep ReLU networks. Throughout the experiments we extensively vary different hyperparameters (e.g., the learning rate, weight decay, and the batch size) and study their effect on the rank of the various matrices in the

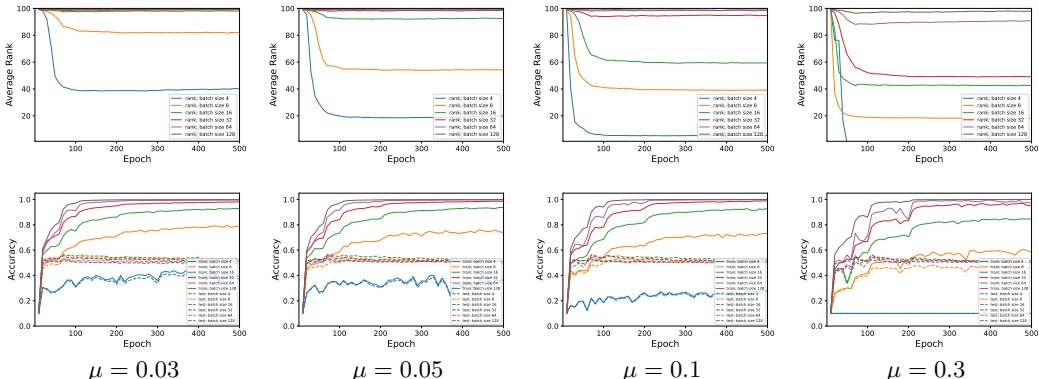

Figure 1: **Average ranks and accuracy rates of MLP-BN-10-100 trained on CIFAR10 with various batch sizes.** The top row shows the average rank across layers, while the bottom row shows the train and test accuracy rates for each setting. Weight decay $\lambda = 5e-4$ was used to train each model. To calculate the rank, we used an $\epsilon = 0.001$ threshold.

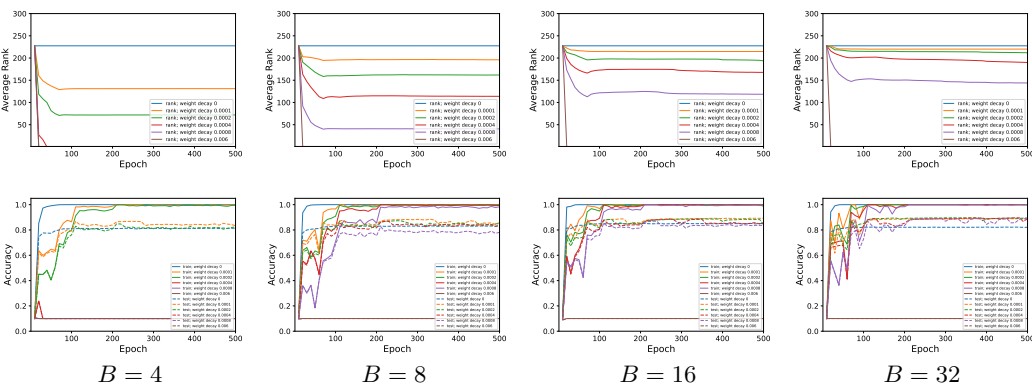

Figure 2: **Average ranks and accuracy rates of ResNet-18 trained on CIFAR10 with varying weight decay.** In this experiment: $\mu = 1.5$ and $\epsilon = 0.001$.

network. In addition, we experimentally analyze the convergence (and non-convergence) of training neural networks with SGD[1].

## 4.1 SETUP

**Evaluation process.** We consider $k$-class classification problems and train a multilayered neural network $f_W : \mathbb{R}^n \to \mathbb{R}^k$ on some balanced training dataset $S$. The model is trained using CE/ MSE loss minimization between its logits and the one-hot encodings of the labels. After each epoch, we compute the averaged rank across the network's weight matrices and its train and test accuracy rates. For a convolutional layer $C^{ij}$, we use $W^{ij}$ as its weight matrix. To estimate the rank of a given matrix $M$, we count how many of the singular values of $M/\|M\|_2$ are $\notin [-\epsilon, \epsilon]$, where $\epsilon$ is a small tolerance value. In these experiments we consider the MNIST and CIFAR10 datasets.

**Architectures.** We consider several network architectures. **(i)** The first architecture is an MLP, denoted by MLP-BN-$L$-$H$, which consists of $L$ hidden layers, where each layer contains a fully-connected layer of width $H$, followed by batch normalization and ReLU activations. On top of that, we compose a fully-connected output layer. **(ii)** The second architecture, denoted by RES-BN-$L$-$H$, consists of a linear layer of width $H$, followed by $L$ residual blocks, ending with a fully-connected layer. Each block computes a function of the form $z + \sigma(n_2(W_2\sigma(n_1(W_1z))))$, where

---

[1]The plots are best viewed when zooming into the pictures.

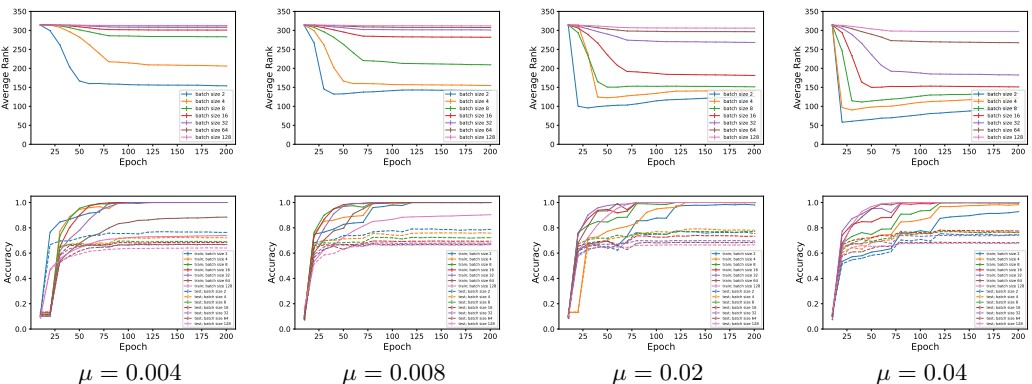

Figure 3: **Average ranks and accuracy rates of ViT trained on CIFAR10 with various batch sizes.** In this experiment: $\lambda = 5\mathrm{e}{-4}$ and $\epsilon = 0.01$.

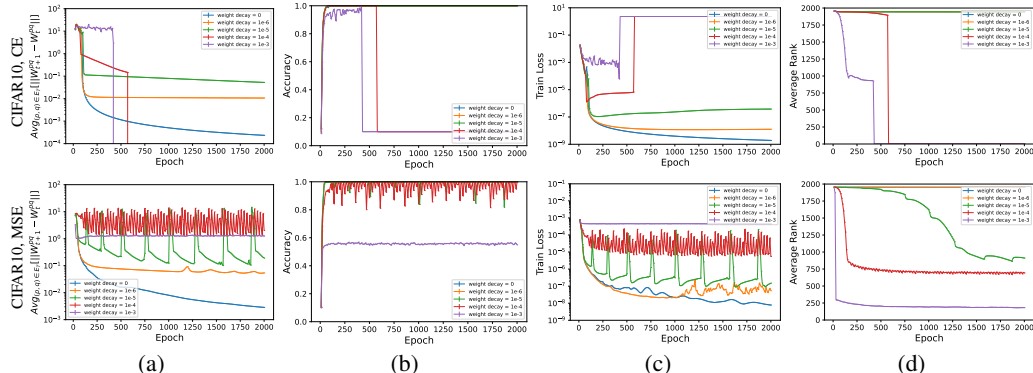

Figure 4: **Convergence of MLP-5-2000 trained on CIFAR10.** In **(a)** we plot the averaged distance between the weight matrices at epoch $t$ and epoch $t+1$, captured by, $d(W_{t+1}, W_t)$. In **(b)** we plot the train accuracy rates, in **(c)** we plot the averaged train loss and in **(d)** we plot the average rank across the trainable matrices.

$W_1, W_2 \in \mathbb{R}^{H \times H}$, $n_1, n_2$ are batch normalization layers and $\sigma$ is the ReLU function. We denote by MLP-$L$-$H$ and RES-$L$-$H$ the same architectures without applying batch normalization. **(iii)** The third, denoted VGG-16, is the convolutional network proposed by Simonyan & Zisserman (2014), but with dropout replaced by batch normalization layers and with only one fully-connected layer at the end. **(iv)** The fourth architecture is the residual network proposed in (He et al., 2016), denoted ResNet-18. **(v)** The fifth layer is a small visual transformer (Dosovitskiy et al., 2020), denoted by ViT. Our ViT splits the input images into patches of size $4 \times 4$ includes 8 self-attention heads, where each one of them consists of 6 self-attention layers. The self-attention layers are followed by two fully-connected layers with dropout probability 0.1 and a GELU activation in between them.

### 4.2 EXPERIMENTS ON RANK MINIMIZATION

In each experiment we trained various models while varying one hyperparameter (e.g., batch size) and leaving the other hyperparameters constant. The models were trained with SGD for cross-entropy loss minimization along with weight decay. For MLP-BN-10-100, ResNet-18 and VGG-16, we decayed the learning rate three times by a factor of $0.1$ at epochs 60, 100, and 200 and training is stopped after 500 epochs. We train instances of ViT using SGD and the the learning rate is decayed by a factor of $0.2$ three times at epochs 60, 100 and training is stopped after 200 epochs. By default, we trained the models with weight decay $\lambda = 5\mathrm{e}{-4}$.

As can be seen in Figs. 1 and 3 by decreasing the batch size, we essentially strengthen the low-rank constraint over the network's matrices, which eventually leads to matrices of lower ranks. This

is consistent with the prediction made in Sec. 3.1 that we learn matrices with lower ranks when training the network with smaller batch sizes. Interestingly, we also notice a regularizing effect for the learning rate; the average rank tends to decrease when increasing the learning rate.

As can be seen in Fig. 2, by increasing $\lambda$ we typically impose stronger rank minimization constraints. Interestingly, it appears that the batch size have little effect on the ranks of the weight matrices when training with $\lambda = 0$ which is exactly the case when our bound is infinite. This empirically validates that weight decay is a necessary to obtain a significant low-rank bias.

### 4.3 EXPERIMENTS ON SGD NOISE

In Sec. 3.2 we showed that convergence to a non-zero function is impossible when training a fully-connected neural network with SGD and weight decay. To validate this prediction, we trained MLP-5-2000 instances examined their convergence as a function of $\lambda$. Each model was trained for CIFAR10 classification using SGD with batch size 128 and learning rate 0.1 for 2000 epochs.

To investigate the convergence of the networks, we measure the averaged distance between the network's matrices between consecutive epochs: $d(W_{t+1}, W_t) := \frac{1}{|E_T|} \sum_{(p,q) \in E_T} \|W_{t+1}^{ij} - W_t^{ij}\|$, where $\{W_t^{ij}\}_{(p,q) \in E_T}$ are the various trainable matrices in the network at epoch $t$. As mentioned in Sec. 3, convergence is possible only when $\lim_{t \to \infty} d(W_{t+1}, W_t) = 0$.

In Fig. 4 we monitor **(a)** $d(W_{t+1}, W_t)$, **(b)** the train accuracy rates, **(c)** the train losses and **(d)** the averaged rank of the trainable matrices. As predicted in Prop. 1, when training with $\lambda > 0$, $W_t$ either converges to zero and $f_{W_t}$ to the zero function (e.g., see the results with $\lambda = 1e-4, 1e-3$) or $W_t$ does not converge (i.e., $d(W_{t+1}, W_t)$ does not tend to zero). Furthermore, we observe that for $\lambda = 0$, $d(W_{t+1}, W_t)$ is smaller by orders of magnitude compared to using $\lambda > 0$ (except for cases when the selection of $\lambda > 0$ leads to $W_t^{ij} \to 0$). Interestingly, even though for certain values of $\lambda > 0$, the training loss and accuracy converged, the network's parameters do not converge.

Finally, when training for cross-entropy loss minimization without weight decay we encounter the 'almost convergence' regime discussed in Sec. 3.2. Namely, even though perfect convergence is impossible, the term $d(W_{t+1}, W_t)$ may become as small as we wish by increasing the size of the neural network. Therefore, since the MLP-5-2000 is relatively large (compared to the dataset's size), we may inaccurately get the impression that $d(W_{t+1}, W_t)$ tends to zero.

## 5 CONCLUSIONS

A mathematical characterization of the biases associated with SGD-trained neural networks is regarded as a significant open problem in the theory of deep learning (Neyshabur et al., 2017). In addition to its independent interest, a low-rank bias – though probably not necessary for generalization – may be a key ingredient in an eventual characterization of the generalization properties of deep networks. In fact, recent results (Huh et al., 2022) and our preliminary experiments (see Figs. 19-20 in the appendix) suggest that low-rank bias in neural networks improves generalization.

By investigating the "SGD Near-Convergence Regime", we proved that SGD together with weight decay induces a low-rank bias in a variety of network architectures. Our result also shows that the batch size used by SGD influences the rank of the learned matrices. This means that the batch size plays an active role in regularizing the learned function.We also prove that when training a fully-connected neural network, SGD noise must always be present, even asymptotically, regardless of batch size, as long as weight decay is used. Weight decay may not be strictly necessary for SGD noise and low-rank bias to appear. This is the case, for example, when training with exponential-type loss functions and Weight Normalization (Salimans & Kingma, 2016).

We hope that our work will spark further research into the near-convergence regime. For instance, it may provide an interesting algorithmic approach to the long-standing problem of developing low-rank regularizers during optimization. It would also be interesting to study whether additional structures (e.g., neural collapse (Papyan et al., 2020), sparsity) emerge during the near-convergence regime and to extend our analysis to more sophisticated learning algorithms (e.g., Adam (Kingma & Ba, 2015)) and learning settings (e.g., unsupervised learning, self-supervised learning, etc).

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

# A    PROOFS

**Lemma 1.** *Let $f_W$ be a neural network and let $C^{ij}$ be a convolutional layer within $f_W$ with parameters matrix $W^{ij}$. Then,* $\operatorname{rank}\left(\nabla_{W^{ij}} f_W(x)\right) \leq N^{ij}$.

*Proof.* Let $x \in \mathbb{R}^{c_1 \times h_1 \times w_1}$ be an input tensor and $C^{ij}$ be a certain convolutional layer with kernel size $(k_1, k_2)$, stride $s$ and padding $p$. We would like to show that $\operatorname{rank}\left(\nabla_{W^{ij}} f_W(x)\right) \leq N^{ij}$. We begin by writing the output of $f_W$ as a sum over paths that pass through $C^{ij}$ and paths that do not. We note that the output can be written as follows,

$$f_W(x) \;=\; \sum_{l_1 \in \operatorname{pred}(l_0)} C^{l_0 l_1} \circ v_{l_1}(x),$$

where $l_0 = L$ and $C^{l_0 l_1} \circ z := C^{l_0 l_1}(z)$. In addition, each layer $v_l$ can also be written as

$$v_{l_1}(x) \;=\; D_{l_1} \odot \sum_{l_2 \in \operatorname{pred}(l_1)} C^{l_1 l_2} \circ v_{l_2}(x),$$

where $D_l := D_l(x) := \sigma'(v_l(x)) \in \mathbb{R}^{c_l \times h_l \times w_l}$.

A path $\pi$ within the network's graph $G$ is a sequence $\pi = (\pi_0, \dots, \pi_T)$, where $\pi_0 = 1$, $\pi_T = L$ and for all $i = 0, \dots, T-1 : \ (v_{\pi_i}, v_{\pi_{i+1}}) \in E$. We can write $f_W(x)$ as the sum of matrix multiplications along paths $\pi$ from $v_1$ to $v_{l_0}$. Specifically, we can write $f_W(x)$ as a follows

$$\begin{aligned}
f_W(x) \;=\;\; & \sum_{\pi \text{ from } i \text{ to } l_0} C^{\pi_T \pi_{T-1}} \circ D_{\pi_{T-1}} \cdots D_{\pi_2} \odot C^{\pi_2 \pi_1} \circ D_{\pi_1} \odot C^{ij} \circ v_j(x) \\
&+ \sum_{\substack{\pi \text{ from } 1 \text{ to } l_0 \\ (i,j) \notin \pi}} C^{\pi_T \pi_{T-1}} \circ D_{\pi_{T-1}} \odot C^{\pi_{T-1} \pi_{T-2}} \cdots D_{\pi_2} \odot C^{\pi_2 \pi_1} \circ x, \\
=:\;\; & A_W(x) + B_W(x)
\end{aligned}$$

where $T = T(\pi)$ denotes the length of the path $\pi$. Since $\sigma$ is a piece-wise linear function with a finite number of pieces, for any $x \in \mathbb{R}^{c_1 \times h_1 \times w_1}$, with measure 1 over $W$, the matrices $\{D_l(x)\}_{l=1}^{L-1}$ are constant in the neighborhood of $W$. Furthermore, $W^{ij}$ does not appear in the multiplications along the paths $\pi$ from 1 to $l_0$ that exclude $(i, j)$. Therefore, we conclude that $\frac{\partial B_W(x)}{\partial W^{ij}} = 0$.

As a next step we would like to analyze the rank of $\frac{\partial A_W(x)}{\partial W^{ij}}$. For this purpose, we rewrite the convolutional layers and the multiplications by the matrices $D_l(x)$ as matrix multiplications.

**Representing $C^{ij}$.** We begin by representing the layer $C^{ij}$ as a linear transformation of its input with $N^{ij}$ blocks of $W^{ij}$. For this purpose, we define a representation of a given 3-dimensional tensor input $z \in \mathbb{R}^{c_j \times h_j \times w_j}$ as a vector $\operatorname{vec}^{ij}(z) \in \mathbb{R}^{N^{ij} c_j k_1 k_2}$. First, we pad $z$ with $p$ rows and columns of zeros and obtain $\operatorname{Pad}_p(z)$. We then vectorize each one of its patches (of dimensions $c_j \times k_1 \times k_2$) that the convolutional layer is acting upon (potentially overlapping) and concatenate them. We can write the vectorized output of the convolutional layer as $U^{ij} \operatorname{vec}^{ij}(z)$, where

$$U^{ij} := \begin{pmatrix}
W^{ij} & \begin{matrix} & & & \\ 0 & 0 & 0 & 0 \end{matrix} & \\
\begin{matrix} 0 & 0 \\ 0 & 0 \end{matrix} & \ddots & \begin{matrix} 0 & 0 \\ 0 & 0 \end{matrix} \\
\begin{matrix} 0 & 0 & 0 & 0 \\ 0 & 0 & 0 & 0 \end{matrix} & & W^{ij}
\end{pmatrix} \tag{7}$$

is a $(N^{ij} c_i) \times (N^{ij} c_j k_1 k_2)$ matrix with $N^{ij}$ copies of $V^{ij}$. We note that this is a non-standard representation of the convolutional layer's operation as a linear transformation. Typically, we write the convolutional layer as a linear transformation $W^{ij}$ acting on the vectorized version $\operatorname{vec}(z) \in \mathbb{R}^{c_j k_1 k_2}$ of its input $z$. Since $\operatorname{vec}^{ij}(z)$ consists of the same variables as in $\operatorname{vec}(z)$ with potentially duplicate items, there is a linear transformation that translates $\operatorname{vec}(z)$ into $\operatorname{vec}^{ij}(z)$. Therefore, we can simply write $V^{ij} \operatorname{vec}(z) = U^{ij} \operatorname{vec}^{ij}(z)$.

**Representing convolutional layers.** Except of $C^{ij}$, we represent each one of the network's convolutional layers $C^{ld}$ in $f_W$ as linear transformations. As mentioned earlier, we can write $\mathrm{vec}(C^{ld}(z)) = V^{ld}\mathrm{vec}(z)$, for any input $z \in \mathbb{R}^{c_d \times h_d \times w_d}$.

**Representing pooling and rearrangement layers.** An average pooling layer or a rearrangement layer $C^{ld}$ can be easily represented as a (non-trainable) linear transformation of its input. Namely, we can write $\mathrm{vec}(C^{ld}(z)) = V^{ld}\mathrm{vec}(z)$ for some constant matrix $V^{ld}$. A max-pooling layer can be written as a composition of ReLU activations and multiple (non-trainable) linear transformations, since $\max(x, y) = \sigma(x - y) + y$. Therefore, without loss of generality we can replace the pooling layers with non-trainable linear transformations and ReLU activations.

**Computing the rank.** Finally, we note that $\mathrm{vec}(C^{ij} \circ z) = U^{ij}\mathrm{vec}^{ij}(z) = V^{ij}\mathrm{vec}(z)$, $\mathrm{vec}(D_l \odot z) = P_l \cdot \mathrm{vec}(z)$ for $P_l := \mathrm{diag}(\mathrm{vec}(D_l))$. Therefore, we can write

$$A_W(x) = \sum_{\pi \text{ from } i \text{ to } l_0} W^{\pi_T \pi_{T-1}} \cdot P_{\pi_{T-1}} \cdots P_{\pi_2} \cdot W^{\pi_2 \pi_1} \cdot P_{\pi_1} \cdot U^{ij} \cdot \mathrm{vec}^{ij}(v_j(x))$$

$$=: a(x)^\top \cdot U^{ij} \cdot b(x),$$

where $a(x)^\top := \sum_{\pi \text{ from } i \text{ to } l_0} W^{\pi_T \pi_{T-1}} \cdot P_{\pi_{T-1}} \cdots P_{\pi_2} \cdot W^{\pi_2 \pi_1} \cdot P_{\pi_1}$ and $b(x) := \mathrm{vec}^{ij}(v_j(x))$. We note that with measure 1, the matrices $\{P_l\}_{l=1}^{L-1}$ are constant in the neighborhood of $W$. In addition, $a(x)$ and $b(x)$ are computed as multiplications of matrices $W^{ld}$ and $P_l$ excluding $(i, j) = (p, q)$. Therefore, with measure 1 over the selection of $W$, the Jacobians of $a(x)$ and $b(x)$ with respect to $V^{ij}$ are 0. Furthermore, due to equation 7 and the definition of $U^{ij}$, we can write

$$a(x)^\top \cdot U^{ij} \cdot b(x) = \sum_{t=1}^{N^{ij}} a_t(x)^\top \cdot V^{ij} \cdot b_t(x),$$

where $a_t(x)$ and $b_t(x)$ are the slices of $a(x)$ and $b(x)$ that are multiplied by the $t$'th $V^{ij}$ block in $U^{ij}$. Since the Jacobians of $a_i(x)$ and $b_i(x)$ with respect to $V^{ij}$ are 0 with measure 1 over the selection of $W$, we have,

$$\frac{\partial a(x)^\top \cdot U^{ij} \cdot b(x)}{\partial V^{ij}} = \sum_{t=1}^{N^{ij}} a_t(x) \cdot b_t(x)^\top. \tag{8}$$

Therefore, we conclude that, with measure 1 over the selection of $W$, we have $\frac{\partial f_W(x)}{\partial V^{ij}} = \sum_{t=1}^{N^{ij}} a_t(x) \cdot b_t(x)^\top$ which is a matrix of rank $\leq N^{ij}$. $\qquad \square$

**Proposition 1** ($\lambda > 0$)**.** *Let $\ell(a, b)$ be a differentiable loss function, $\lambda > 0$, and let $f_W(x)$ be a ReLU neural network, where $\mathrm{succ}(1) = \{p\}$ and $(v_p, v_1) \in E_T$. Let $\{x_i^k\}_{k=1}^{N^{p1}}$ be the $N^{p1}$ patches of $x_i$ used by the layer $C^{p1}$. Let $W$ be a convergence point of mini-batch SGD for minimizing $L_S^\lambda(f_W)$ (see equation 4). Then, either $f_W \equiv 0$ or $\forall i, j \in [m] : \{x_i^k, x_j^k\}_{k=1}^{N^{p1}}$ are linearly dependent tensors.*

*Proof.* Since $\{(v_p, v_1) \in E \mid v_p \in V\} \subset E_T$ is of size 1, we denote this single layer by $(v_p, v_1)$. Following the proof of Lem. 1, we define a representation of a given 3-dimensional tensor input $x \in \mathbb{R}^{c_j \times h_j \times w_j}$ as a vector $\mathrm{vec}^{ij}(x) = (x^1 \| \ldots \| x^{N^{ij}}) \in \mathbb{R}^{N^{ij} c_j k_1 k_2}$, where $x^k$ is the vectorization of the $k$'th $c_j \times k_1 \times k_2$ patch of $x$. Similar to the proof of Lem. 1, we can write

$$f_W(x) = \sum_{\pi \text{ from } 1 \text{ to } L} C^{\pi_T \pi_{T-1}} \circ D_{\pi_{T-1}} \odot C^{\pi_{T-1} \pi_{T-2}} \cdots D_{\pi_2} \odot C^{\pi_2 \pi_1} \circ x$$

$$= H(x) \cdot U^{p1} \cdot \mathrm{vec}^{p1}(x) \tag{9}$$

where $H(x) := \sum_{\pi \text{ from } p \text{ to } L} V^{\pi_T \pi_{T-1}} \cdot P_{\pi_{T-1}}(x_i) \cdots V^{\pi_2 \pi_1} \cdot P_{\pi_1}(x_i)$ and denote $H_i := H(x_i)$ and $H_i = (H_i^1, \ldots, H_i^{N^{p1}})$, where $H_i^k$ is of dimension $c_1 k_1 k_2$. Hence, we can write,

$$\frac{\partial f_W(x_i)}{\partial W^{p1}} = \sum_{k=1}^{N^{p1}} H_i^k \cdot (x_i^k)^\top.$$

We would like to show that $\{x_i^k, x_j^k\}_{k=1}^{N^{p1}}$ are linearly dependent vectors or $f_W \equiv 0$. Assume the opposite by contradiction, i.e., that $\{x_i^k, x_j^k\}_{k=1}^{N^{p1}}$ are not linearly dependent and that $f_W \not\equiv 0$. In particular, by equation 9, we have, $W^{p1} \neq 0$.

According to the analysis in Sec. 3.2, $\{\frac{\partial f_W(x_i)}{\partial \text{vec}(W^{p1})}\}_{i=1}^m$ are collinear vectors. Therefore, for any pair $i, j \in [m]$, there is a scalar $\alpha_{ij}^p \in \mathbb{R}$, such that, $\sum_{k=1}^{N^{p1}} H_i^k \cdot (x_i^k)^\top = \alpha_{ij} \sum_{k=1}^{N^{p1}} H_j^k \cdot (x_j^k)^\top$.

Consider a given index $i \in [m]$. We would like to show that either $H_i = 0$ or $\{x_i^k, x_j^k\}_{k=1}^{N^{p1}}$ is a set of linearly dependent vectors. Assume that $H_i = 0$. Then, $\frac{\partial f_W(x_i)}{\partial W^{p1}} = 0$ and $M^{p1} = 0$ and $W^{p1} = 0$ according to equation 6, which implies that $f_W \equiv 0$ (since $(v_p, v_1)$ is the only connection starting from $v_1$). Assume that $H_i \neq 0$. Then, there exist $k \in [N^{p1}]$ and $r \in [\dim(H_i)/N^{p1}]$, for which the $r$'th coordinate of $H_i^k$ is non-zero. Therefore, the $r$'th row of $\sum_{k=1}^{N^{p1}} H_i^k \cdot (x_i^k)^\top - \alpha_{ij} \sum_{k=1}^{N^{p1}} H_j^k \cdot (x_j^k)^\top = 0$ is a non-trivial linear combination of the vectors $\{x_i^k, x_j^k\}_{k=1}^{N^{p1}}$.

$\square$

**Proposition 2** ($\lambda = 0$). *Let $\lambda = 0$ and let $\ell$ be a differentiable loss function. Let $f_W(x)$ be a ReLU neural network, where $\text{succ}(1) = \{p\}$ and $(v_p, v_1) \in E_T$ is fully-connected. Let $\{x_i^k\}_{k=1}^{N^{p1}}$ be the $N^{p1}$ patches of $x_i$ used by the layer $C^{p1}$. Let $W$ be a convergence point of mini-batch SGD for minimizing $L_S^\lambda(f_W)$. Then, $\forall\, i \in [m] : \frac{\partial \ell(f_W(x_i), y_i)}{\partial f_W(x_i)} = 0$ or $f_W(x_j) = 0$ or $\forall\, j \in [m] : \{x_i^k, x_j^k\}_{k=1}^{N^{p1}}$ are linearly dependent tensors.*

*Proof.* Let $i \in [m]$ be an index for which $\frac{\partial \ell(f_W(x_i), y_i)}{\partial f_W(x_i)} \neq 0$. Then, by equation 5, for all $j \in [m]$, we have

$$\frac{\partial f_W(x_i)}{\partial W^{p1}} = \left[\frac{\partial \ell(f_W(x_j), y_j)}{\partial f_W(x_j)}\right] / \left[\frac{\partial \ell(f_W(x_i), y_i)}{\partial f_W(x_i)}\right]^{-1} \cdot \frac{\partial f_W(x_j)}{\partial W^{p1}}. \tag{10}$$

In particular, $\frac{\partial f_W(x_i)}{\partial \text{vec}(W^{p1})}$ and $\frac{\partial f_W(x_j)}{\partial \text{vec}(W^{p1})}$ are collinear vectors (for all $j \in [m]$). Hence, by the proof of Prop. 1, either $f_W(x_i) = 0$ or $\{x_i^k, x_j^k\}_{i=1}^m$ is a set of linearly dependent vectors.

$\square$

## B ADDITIONAL EXPERIMENTS

**Experiments on rank minimization.** To further demonstrate the bias towards rank minimization of SGD with weight decay, we conducted a series of experiments with different learning settings. We follow the same training and evaluation protocol described in Sec. 4.2. The results are summarized in Figs. 6-18.

**Experiments on SGD noise.** We repeated the experiment in Fig. 4 for training the models on MNIST. As can be seen in Fig. 5, similar to the previous experiment, when the models were trained with $\lambda > 0$, the weights $W_t$ were not able to converge (i.e., $d(W_{t+1}, W_t)$ does not tend to zero), even though for certain values of $\lambda > 0$ the training accuracy and loss converged. On the other hand, when $\lambda = 0$, the distance $d(W_{t+1}, W_t)$ tends to zero.

**Low-rank bias and generalization.** We looked into the connection between low-rank bias and generalization. In Figs. 19-20 we trained ResNet-18 and VGG-16 instances on CIFAR10 while varying the batch size and keeping $\lambda$ and $\mu$ constant. To provide a fair comparison, we chose $\lambda$ and $\mu$ in each setting to ensure that all models fit the training data perfectly. As can be seen, models trained with smaller batch sizes, i.e. models with lower rank in their weights, tend to generalize better. Based on these findings, we hypothesize that when two neural networks of the same architecture are trained with SGD with different hyperparameters and perfectly fit the data, the one with a lower average rank will outperform the other at test time.

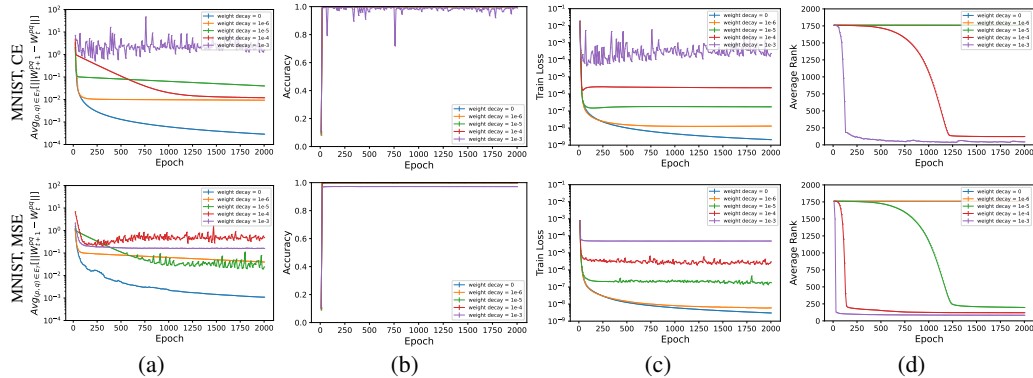

Figure 5: **Convergence of MLP-5-2000 trained on MNIST with CE/ MSE loss.** In **(a)** we plot the averaged distance between the weight matrices at epoch $t$ and epoch $t + 1$, captured by, $\frac{1}{|E_T|} \sum_{(p,q) \in E_T} \|W_{t+1}^{ij} - W_t^{ij}\|$. In **(b)** we plot the train accuracy rates, in **(c)** we plot the averaged train loss and in **(d)** we plot the average rank across the trainable matrices.

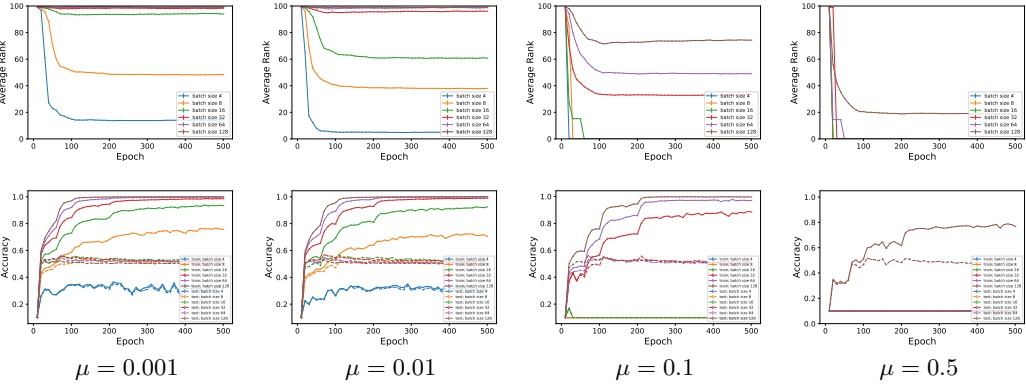

Figure 6: **Average rank of MLP-BN-10-100 trained on CIFAR10 with various batch sizes.** Each model was trained with a $\lambda = 5e-4$ weight decay. To estimate the rank, we used an $\epsilon = 0.001$ threshold.

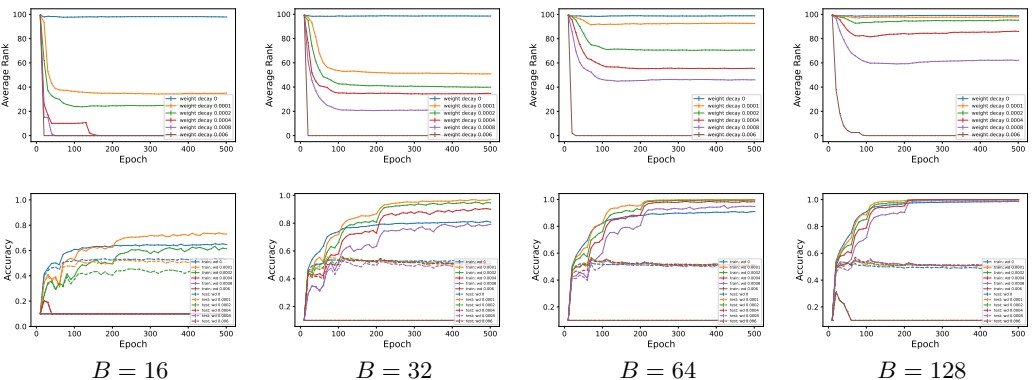

Figure 7: **Average rank of MLP-BN-10-100 trained on CIFAR10 with varying** $\lambda$**.** Each model was trained with a $\mu = 0.1$ initial learning rate and $0.9$ momentum. To estimate the rank, we used an $\epsilon = 0.001$ threshold.

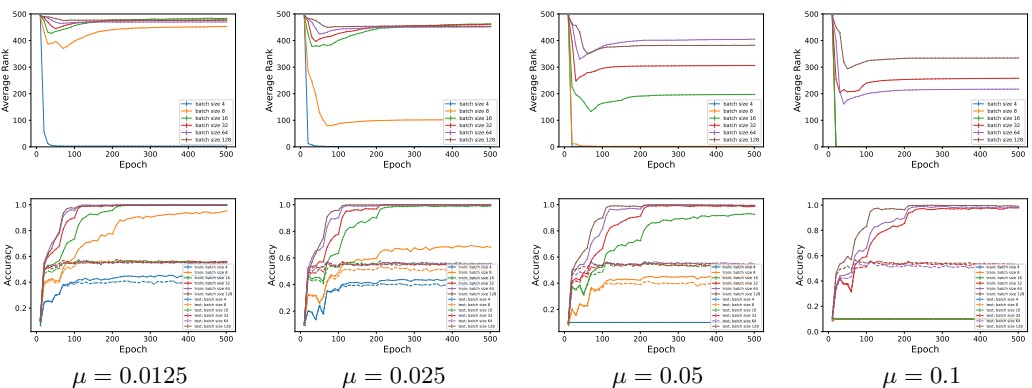

Figure 8: **Average rank of RES-BN-5-500 trained on CIFAR10 with various batch sizes.** Each model was trained with a $\lambda = 5e-4$ weight decay. To estimate the rank, we used an $\epsilon = 0.001$ threshold.

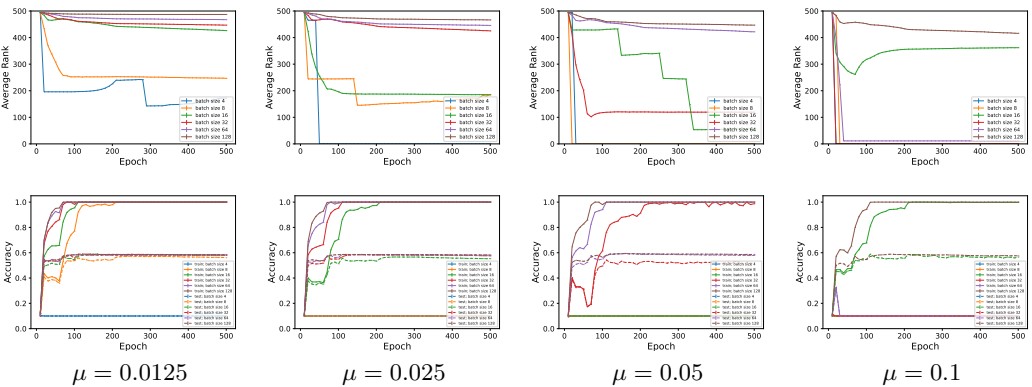

Figure 9: **Average ranks and accuracy rates of MLP-5-500 trained on CIFAR10 with various batch sizes.** Each model was trained with a $\lambda = 5e-4$ weight decay. To estimate the rank, we used an $\epsilon = 0.001$ threshold.

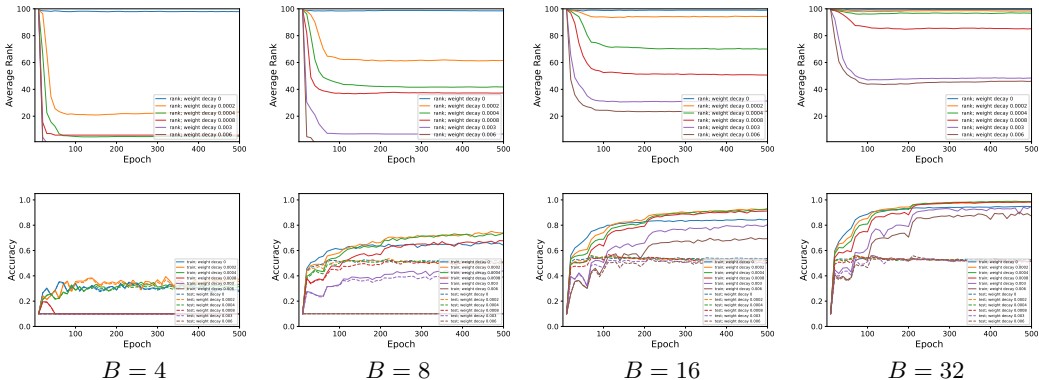

Figure 10: **Average ranks and accuracy rates of MLP-BN-10-100 trained on CIFAR10 with varying** $\lambda$. Each model was trained with a $\mu = 0.1$ initial learning rate. To estimate the rank, we used an $\epsilon = 0.001$ threshold.

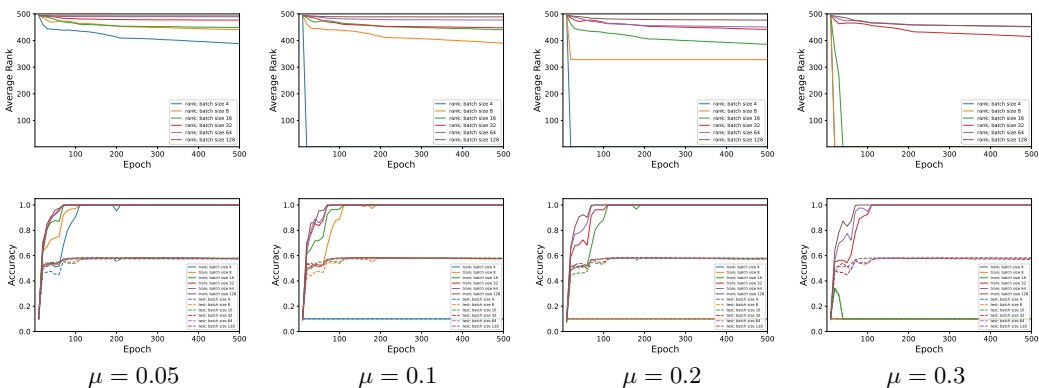

Figure 11: **Average ranks and accuracy rates of MLP-5-500 trained on CIFAR10 with various batch sizes.** The models were trained with a $\lambda = 5\mathrm{e}{-}4$ weight decay. To estimate the rank, we used an $\epsilon = 0.001$ threshold.

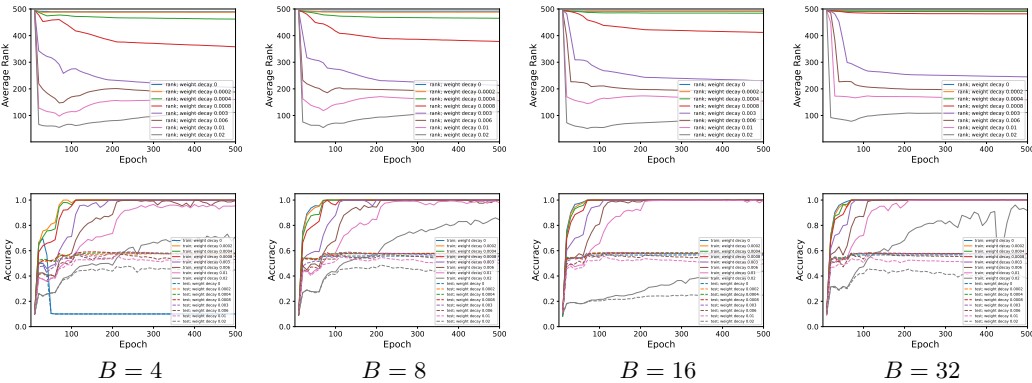

Figure 12: **Average ranks and accuracy rates of MLP-5-500 trained on CIFAR10 with varying** $\lambda$. The models were trained with a $\mu = 0.025$ initial learning rate. To estimate the rank, we used an $\epsilon = 0.001$ threshold.

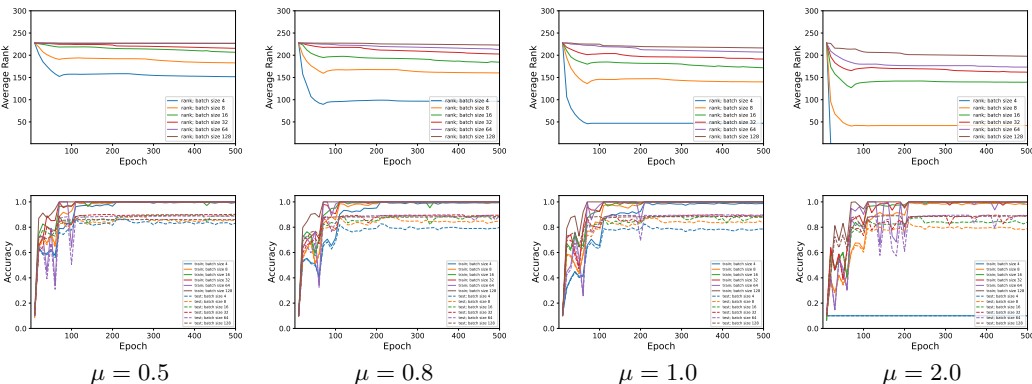

Figure 13: **Average ranks and accuracy rates of ResNet-18 trained on CIFAR10 with various batch sizes.** The models were trained with a $\lambda = 5\mathrm{e}{-4}$ weight decay. To estimate the rank, we used an $\epsilon = 0.001$ threshold.

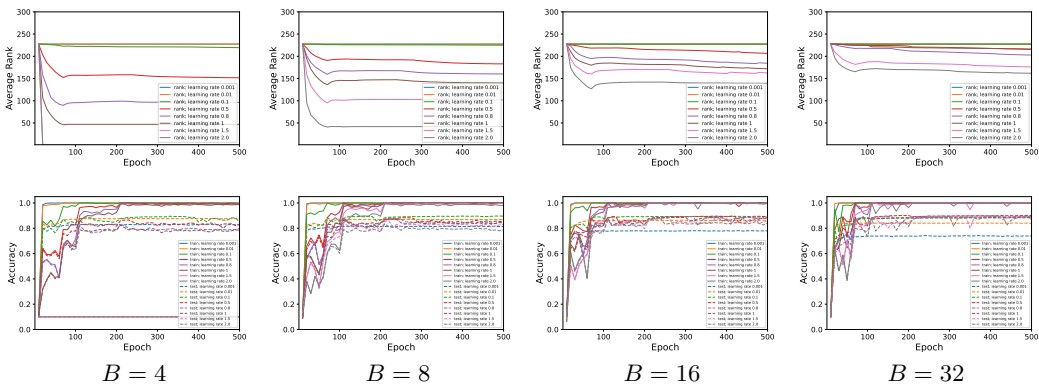

Figure 14: **Average ranks and accuracy rates of ResNet-18 trained on CIFAR10 with varying $\mu$.** The models were trained with a $\mu = 5\mathrm{e}{-4}$ initial learning rate. To estimate the rank, we used an $\epsilon = 0.001$ threshold.

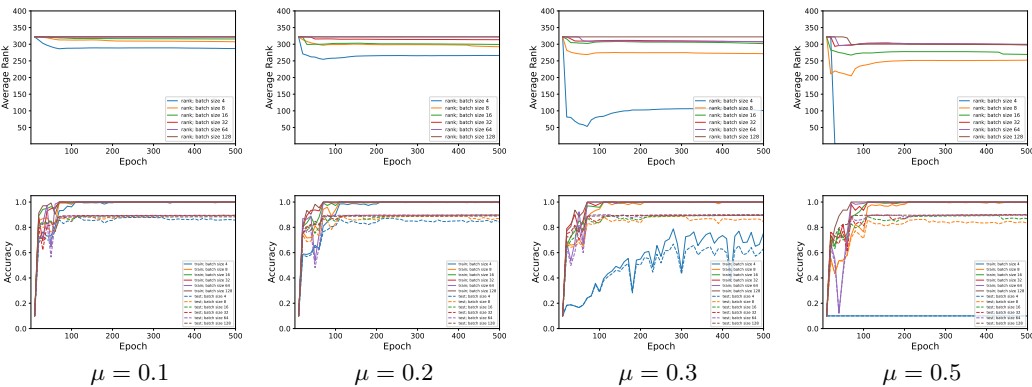

Figure 15: **Average ranks and accuracy rates of VGG-16 trained on CIFAR10 with various batch sizes.** The models were trained with a $\mu = 5\mathrm{e}{-4}$ initial learning rate. To estimate the rank, we used an $\epsilon = 0.001$ threshold.

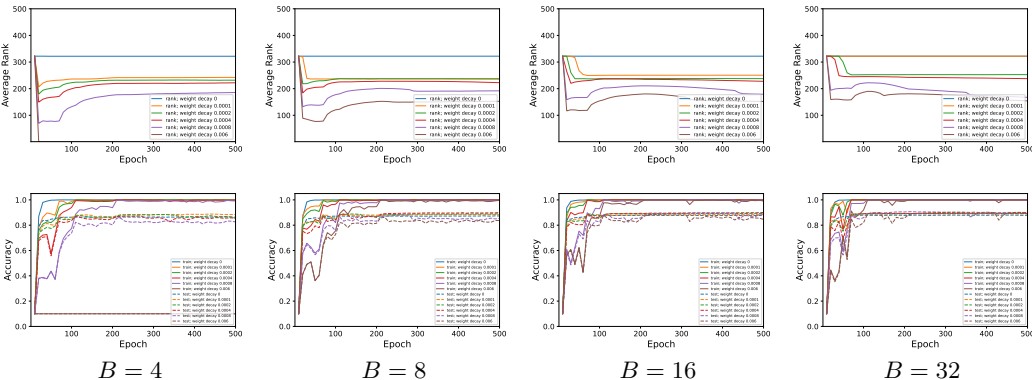

Figure 16: **Average ranks and accuracy rates of VGG-16 trained on CIFAR10 with varying $\lambda$.** The models were trained with a $\mu = 0.1$ initial learning rate. To estimate the rank, we used an $\epsilon = 0.01$ threshold.

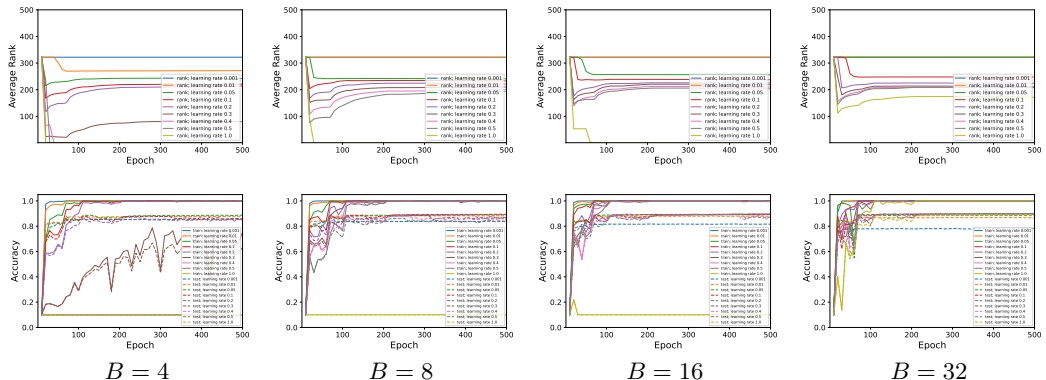

Figure 17: **Average ranks and accuracy rates of VGG-16 trained on CIFAR10 with varying $\mu$.** The models were trained with a $\mu = 5e-4$ weight decay. To estimate the rank, we used an $\epsilon = 0.001$ threshold.

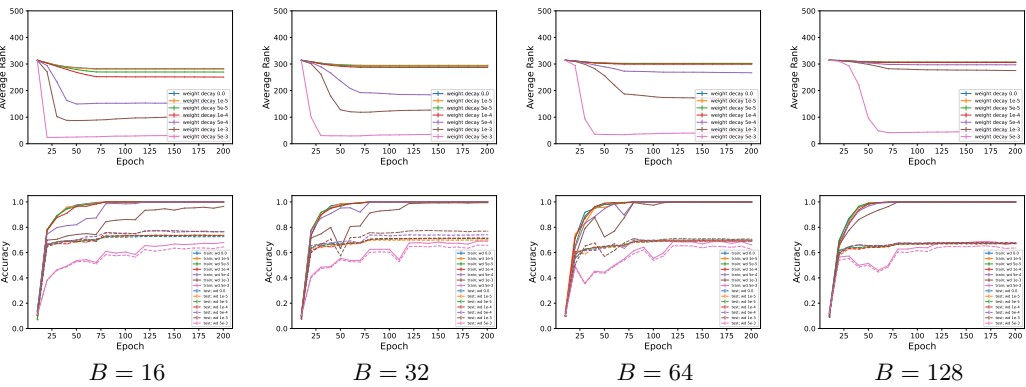

Figure 18: **Average ranks and accuracy rates of ViT trained on CIFAR10 with varying $\lambda$.** The models were trained with a $\mu = 4e-2$ initial learning rate. To estimate the rank, we used an $\epsilon = 0.01$ threshold.

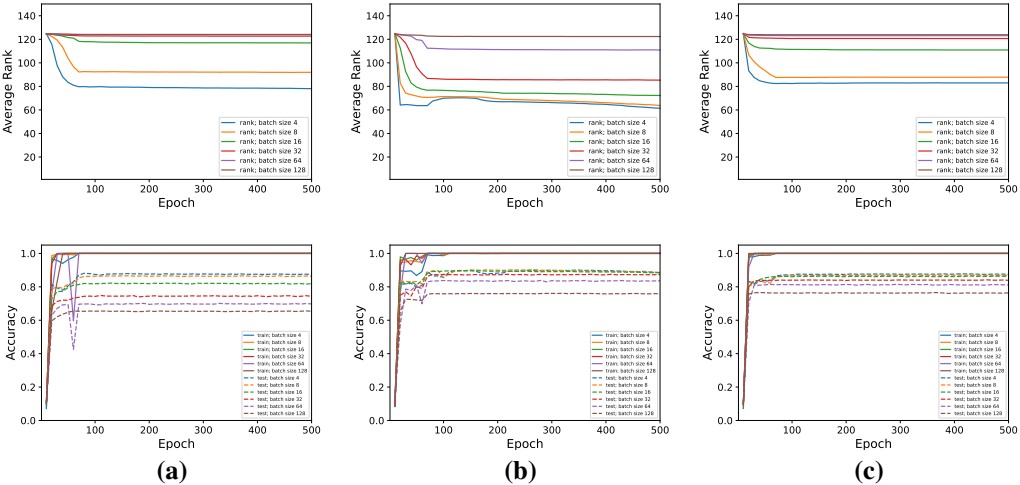

Figure 19: **Average ranks and accuracy rates of ResNet-18 trained on CIFAR10 with different batch sizes.** We show the relationship between average rank and test performance. **(a)** was trained with $\mu = 0.001, \lambda = 6e-3$, **(b)** was trained with $\mu = 0.005, \lambda = 6e-3$, and **(c)** was trained with $\mu = 0.01, \lambda = 4e - 4$. To estimate the rank, we used an $\epsilon = 0.05$ threshold.

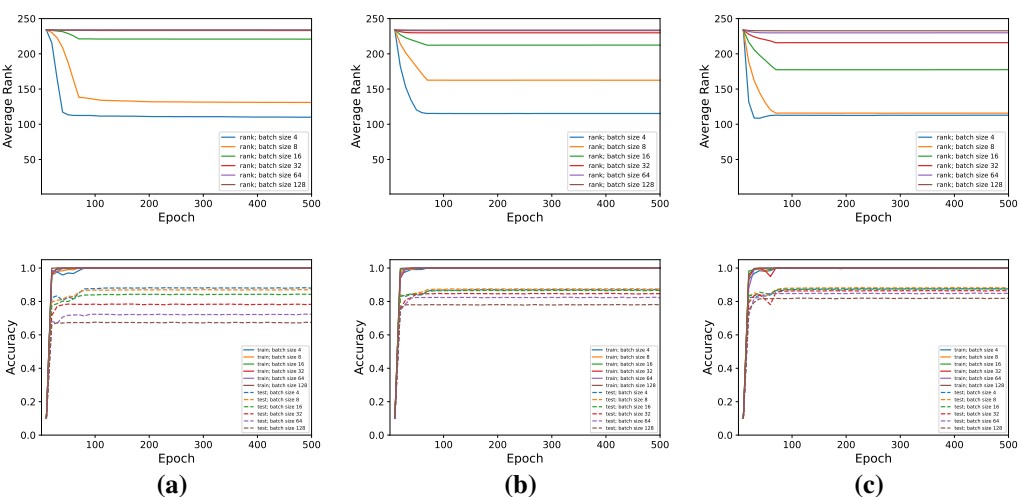

Figure 20: **Average ranks and accuracy rates of VGG-16 trained on CIFAR10 with different batch sizes.** We show the relationship between average rank and test performance. **(a)** was trained with $\mu = 0.001, \lambda = 6e - 3$, **(b)** was trained with $\mu = 0.005, \lambda = 5e - 4$, and **(c)** was trained with $\mu = 0.01, \lambda = 4e - 4$. To estimate the rank, we used an $\epsilon = 0.04$ threshold.

