# OpenReview forum: "SGD and Weight Decay Provably Induce a Low-Rank Bias in Neural Networks"
_ICLR.cc/2023/Conference — Submitted to ICLR 2023_

### Official Review · Reviewer_g7pt · 2022-10-20

**Confidence:** 4
**Clarity, Quality, Novelty And Reproducibility:** I already explained my point above.
**Correctness:** 1
**Technical Novelty And Significance:** 1
**Empirical Novelty And Significance:** 3
**Recommendation:** 1

**Strength And Weaknesses:**

The authors of the paper show that, due to the weight decay, SGD **cannot**, in parameter space, converge towards a point. This is done in Section 3.2 and I totally agree with this result. It is also said that weight decay is crucial to have this property. True.

That being said, I believe that it is easy to further show that the presence of weight decay will cause the non-degeneracy of the noise implying the irreducibility of the SGD as a Markov Chain on a certain domain in parameter space. **Hence**, *the near convergence* depicted in the first equation of Section 3 cannot occur (even in expectation): there is not single point towards which the dynamics can concentrate except if the authors prove it on **the law** of the iterates ! To give a concrete appreciation of this vacuous assumption, consider the Ornstein-Uhlenbeck process: $d W_t = - W_t dt + \sqrt{2} dB_t$, there shall not exist a $W^*$ such that $\mathbb{E}|W_t - W^*| \to 0$ as $t \to \infty$. Here the good object that authors can consider if they want to pursue in this direction is either to focus on the law of $W_t$, either to consider $W_*$ as a random variable coming from this law.

Finally, all the remaining calculations and conclusions collapse.

**Summary Of The Paper:**

The authors of the presented paper try to present why SGD + weight decay tend to converge towards low rank solutions.

**Summary Of The Review:**

The authors consider as a cornerstone of their analysis an empty assumption. Maybe this idea of low rank can be fruitful in the future and I encourage the authors to pursue their efforts but under this shape I will reject the paper.

---

### Official Review · Reviewer_897C · 2022-10-22

**Confidence:** 3
**Correctness:** 2
**Technical Novelty And Significance:** 2
**Empirical Novelty And Significance:** 2
**Recommendation:** 3

**Clarity, Quality, Novelty And Reproducibility:**

As far as I am aware, the results are novel. The paper is well-written and clear, except for the issues discussed above.

**Strength And Weaknesses:**

Let me start with the theoretical results on SGD noise. The main insight here is that when the objective contains $\ell_2$ regularization, the requirement of having zero gradients for all possible batches is very strong, and implies that the gradients of the model for different examples must be collinear. As a result, the authors show that stationary points exist only in degenerate cases. I think that it is a nice and simple insight, and I am not aware of prior works that used it.

Regarding the theoretical result on rank minimization, I must say that I am confused. The authors prove (in Theorem 1) that when training with small batches and weight decay, the distance between the weight matrix and a low-rank matrix is bounded by the gradient of the regularized loss w.r.t. a batch of inputs. Then, the authors claim that they expect this gradient to be small “near convergence”. However, the theoretical results on SGD noise imply that this assumption is strong and might not hold in practice. Thus, it is not clear whether for sufficiently large time $t$ the gradient of the loss w.r.t. a batch should be small. The authors justify this assumption with Eq. (3). However, Eq. (3) follows from the “convergence in mean” assumption, which is puzzling. The “convergence in mean” assumption tells us that there exists $W^*$ such that for a sufficiently large $t$, we have w.h.p. that $\| W_t -W^* \|$ is small. Thus, it assumes that SGD (possibly with a small batch size) behaves in a deterministic way, since it is guaranteed to be close (w.h.p.) to this specific $W^*$. I don’t think that this assumption should hold in practice.
Moreover, in the empirical results in Figure 4(a) it seems that the parameters do not converge when training with weight decay. If the authors claim that the gradient does not converge to zero but is “small” for large $t$, then the question is what can be considered “small”. Thus, it requires further analysis and more evidence.

The experiments are nice, and suggest that small batch sizes and weight decay encourage rank minimization, and also that SGD with weight decay does not converge. However, the paper aims not only to show empirically that there is rank minimization, but also to explain it with Theorem 1. Hence, I would be happy to see how the theoretical bound (the RHS of the equation in Theorem 1) looks in practice (at least roughly). It is not clear to me whether Theorem 1 indeed explains the empirical results on rank minimization.

Additional comments:
- Related work:
    - Razin and Cohen (2020) came before Li et al. (2020), and not “later”.
    - Ji and Telgarsky (2020) did not assume that the networks are univariate.
- It seems that the notation $(v_i,v_j)$ is inconsistent (sometimes denotes connection from j to i and sometimes from i to j). There is also a typo in the definition of $v_i$.
- Proposition 2: $f_W(x_j)$ should be $f_W(x_i)$ ?
- First paragraph: “(iii)”
- Page 8: “the the”

**Summary Of The Paper:**

The paper considers training neural networks with SGD and weight decay. It contains two main theoretical results:
(1) When the batch size is small, convergence of SGD to parameters $W$ implies that the parameters matrices $W^{ij}$ are close to low-rank matrices.
(2) Unless the dataset is degenerate (i.e., patches of each two examples are linearly dependent) or the learned model is trivial, SGD with weight decay cannot converge to a stationary point. SGD without weight decay can converge to a non-trivial stationary point if the dataset is degenerate, or the predictor perfectly fits the labels.

The authors demonstrate empirically that increasing the weight decay or decreasing the batch size encourages small ranks. They also demonstrate empirically that SGD with weight decay does not converge to non-zero functions.

**Summary Of The Review:**

Due to the issues discussed above, I recommend rejection. I would be happy to read the authors’ response and reconsider.

---

### Official Review · Reviewer_g7fW · 2022-10-24

**Confidence:** 3
**Correctness:** 3
**Technical Novelty And Significance:** 2
**Empirical Novelty And Significance:** 2
**Recommendation:** 3

**Clarity, Quality, Novelty And Reproducibility:**

**Clarity**

Good

**Novelty**

To me, the theoretical results do not convey much information, and may not be able to explain the experimental observations.

**Strength And Weaknesses:**

**Strength**

The paper is well-written, and the message is easy to get.

**Weakness**

1. The theoretical results seem to be paradoxical. Theorem 1 approximates the weight matrix by a low-rank matrix when the batch gradient norm is small. However, Proposition 1 shows that this is impossible.

2. Theorem 1 does not necessarily lead to the low-rank bias. First of all, the bound has a $1/2\lambda$ factor. As $\lambda$ is usually small in practice ($5e^{-4}$, as an example), it is quite possible that the bound is vacuous. Secondly, even when the bound is small, it only indicates that the weight matrix can be approximated by a low-rank matrix in terms of $\ell^2$ error. This does not necessarily imply the weight matrix itself is low rank. Finally, even if Theorem 1 indicates a low-rank bias, the results are not comparable across different batch-size -- it only shows that with different batch sizes $B$, the weight matrix can be approximated by matrices with different ranks $R(B)$. It does not rule out the possibility that matrices with ranks smaller than $R(B)$ can also approximate the weight matrices.

3. What does Proposition 1 imply? While SGD does not converge to a stationary point, it can oscillate around it -- which is usually the case in practice.

4. I wonder whether the experiments can be explained by the theoretical results. Specifically, for figure 1 and figure 2, I suggest that the authors should plot the batch gradient norms to see whether they are small.

**Summary Of The Paper:**

This paper studies the convergence of SGD over deep ReLU neural networks.

**Theoretical Results**

In the case with weight decay, the authors prove:

1. The weight matrix can be approximated by a low-rank matrix with error bounded by the smallest batch gradient norm across batches (batch gradient stands for the gradient of the loss calculated from batch).

2.  If all batch gradients are $0$, then all the parameters are $0$.

**Experimental Results**

1.  On image tasks, decreasing the batch size or increasing the learning rate leads to a decrease in weight matrix rank.

2. On image tasks, with weight decay, the change in parameters $W_{t+1}-W_{t}$ does not converge to $0$.

**Summary Of The Review:**

Based on the above evaluations, I think that the current version of this paper needs to be improved to meet the bar of acceptance, especially the theory part.

---

### Official Review · Reviewer_c3Es · 2022-10-31

**Confidence:** 4
**Correctness:** 2
**Technical Novelty And Significance:** 2
**Empirical Novelty And Significance:** 2
**Recommendation:** 3

**Clarity, Quality, Novelty And Reproducibility:**

There is no major problem with the writing of the paper, and the proofs seem correct. However, there is an issue with the hyper-parameter tuning in the experiments, as pointed out above. The theoretical results about SGD noise are novel but not very insightful.


**Strength And Weaknesses:**

### **Theorem 1 and Experiments**

Why are there no experiments showing how small the right-hand side is in theorem 1 for the models considered in the experiments, i.e., is SGD actually in the near-convergence regime? Further, it would be good to discuss in the experiments how small $BN^{ij}$ is for the considered models.

Also, considering the near-convergence regime without mentioning the step size doesn't seem reasonable. As the authors mentioned, with decreasing step size, SGD would indeed converge, albeit forcefully. And with constant step size, one would hope it converges up to a ball around some fixed point, even if the consecutive steps don't imply convergence (see papers like [this](https://arxiv.org/abs/1707.06386) and [this](https://arxiv.org/abs/2006.07904)). Usually, there is a periodic step-size decay when using SGD. Thus the correct notion of convergence should be measured by the actual update (step size times the gradient) normalized by the scale of the parameter.

Further, the bound in the theorem does not show any non-asymptotic dependence on the step size $\mu$ or other problem-dependent parameters (such as the initial parameter norm, lipschitzness, etc.). As a result, it is hard to interpret the result without understanding how small the right side is for a specific neural network, optimizer, and set of hyper-parameters. This gets more confusing when one looks at the experiments. It is unclear if the rank-minimization happens before or after the model has "converged."

Another issue is the conflation between step size and batch size. Ideally, one would tune the step size separately for each batch size, i.e., ensure that the step size offers optimal convergence for the training budget. This would avoid needing four different columns in figures 1 and 3. It seems like some of the differences in rank can be ascribed to non-optimal step-size selection. For instance in figure 1, $B=4, \mu=0.03$ curves look much more comparable to $B=32, \mu=0.3$ curves, than $B=4, \mu=0.3$ (and similarly for figure 3). Based on the theory, there seems to be no reason to compare different batch sizes with the same step size. Am I missing something? If not, the authors should report revised experiments where the step size is treated as a part of the optimizer and tuned optimally.

### **SGD Noise**

The observations in section 3.2 are not very surprising. These are good-to-know results but add little to our understanding of the implicit regularization of SGD. For instance, without weight decay, exponential losses are minimized at infinity. In contrast, the square loss is minimized at a unique point is a classical learning theory fact about surrogate losses. And this is often exploited in studying neural networks in the interpolation regime. Proposition 2 almost seems tautological. And showing proposition 1 is not complicated, given the architecture's formalism. Understanding how the SGD noise goes to zero as the batch size increases would be exciting and might not be a problematic extension.





**Summary Of The Paper:**

This paper proposes studying SGD at the "near-convergence regime," which it defines as a point in training where each subsequent update is small compared to the actual weights. First, the paper shows in Theorem 1 that in this regime, with weight decay $\lambda>0$ and batch size $B$, the parameter matrices are close to a low rank ($\propto B$) approximation (in any matrix norm). Then the paper shows that for $\lambda>0$, SGD doesn't converge for usual non-stylized data sets. This means there is always some non-zero "SGD-noise" when training with weight decay. However, when $\lambda=0$ and the loss function has real minima (such as square loss), SGD converges only if the model perfectly interpolates the data. Experiments are provided to evaluate these theoretical results.

**Summary Of The Review:**

I am concerned about the technical novelty of the paper and thus am inclined towards rejecting it. I believe the results in the paper require more work and need to be better put in context with the existing literature in deep learning theory.

------------------------------------------------------------------------------------------------------------------------------------------------------------

After reading the concerns pointed out by the reviewers, especially reviewer g7pt, I am convinced there are major issues with the results in the paper. Thus I am reducing my score.

---

### Decision · Program_Chairs · 2023-01-20

**Decision:**

Reject

**Justification For Why Not Higher Score:**

Reviewer perspectives and scores clear; no author responses/revision.

**Justification For Why Not Lower Score:**

.

**Metareview: Summary, Strengths And Weaknesses:**

This is a nice paper studying the interplay between weight decay and low rank bias in neural networks. Unfortunately, as identified by the reviewers, there are many limitations of the study, and the authors did not respond. I urge the authors to continue their work and submit to a future venue.